# Hypoxia and Its Influence on Radiotherapy Response of HPV-Positive and HPV-Negative Head and Neck Cancer

**DOI:** 10.3390/cancers13235959

**Published:** 2021-11-26

**Authors:** Marilyn Wegge, Rüveyda Dok, Sandra Nuyts

**Affiliations:** 1Laboratory of Experimental Radiotherapy, Department of Oncology, University of Leuven, 3000 Leuven, Belgium; marilyn.wegge@kuleuven.be (M.W.); ruveyda.dok@kuleuven.be (R.D.); 2Department of Radiation Oncology, Leuven Cancer Institute, UZ Leuven, 3000 Leuven, Belgium

**Keywords:** head and neck squamous cell carcinoma, human papillomavirus, radiotherapy, hypoxia

## Abstract

**Simple Summary:**

HPV-positive HNSCCs are characterized by a different biology and demonstrate better therapy response and survival compared to alcohol/tobacco-related HNSCCs. Although we have a better understanding of the biology of both groups of HNSCC, the biological factors, especially environmental factors associated with the increased radiotherapy response, are still unclear. In this manuscript, we review the effects of an important microenvironmental factor, namely, low oxygen levels, also known as hypoxia, on the radiotherapy response and the tumor biology of HPV-positive and HPV-negative HNSCCs. In addition, we provide an overview of the current strategies to detect and target hypoxia, with a description of important clinical trials.

**Abstract:**

Head and neck squamous cancers are a heterogeneous group of cancers that arise from the upper aerodigestive tract. Etiologically, these tumors are linked to alcohol/tobacco abuse and infections with high-risk human papillomavirus (HPV). HPV-positive HNSCCs are characterized by a different biology and also demonstrate better therapy response and survival compared to alcohol/tobacco-related HNSCCs. Despite this advantageous therapy response and the clear biological differences, all locally advanced HNSCCs are treated with the same chemo-radiotherapy schedules. Although we have a better understanding of the biology of both groups of HNSCC, the biological factors associated with the increased radiotherapy response are still unclear. Hypoxia, i.e., low oxygen levels because of an imbalance between oxygen demand and supply, is an important biological factor associated with radiotherapy response and has been linked with HPV infections. In this review, we discuss the effects of hypoxia on radiotherapy response, on the tumor biology, and the tumor microenvironment of HPV-positive and HPV-negative HNSCCs by pointing out the differences between these two tumor types. In addition, we provide an overview of the current strategies to detect and target hypoxia.

## 1. Head and Neck Squamous Cell Cancer (HNSCC)

HNSCC arises from epithelial cells and occurs in the upper aero-digestive tract. It is the sixth most common cancer worldwide and the incidence is still rising [1]. HNSCC is a heterogeneous group of cancers and can be divided into two distinct tumor types based on their etiology being induced either by alcohol and/or tobacco or an infection with human papillomavirus (HPV). Currently, around 75–85% of head and neck cancers are associated with tobacco and alcohol abuse, which are also referred to as HPV-negative tumors, but the proportion of HPV-positive tumors is increasing worldwide [2]. Over the last decades, it has become clear that these two tumor types are distinct entities with different biological and clinical characteristics.

On the biological level, infection of the squamous epithelium with a high-risk human papillomavirus, mainly HPV-16 and HPV-18, can cause malignant transformation through the action of the two main oncoproteins E6 and E7. The E6 oncoprotein disrupts the p53 pathway, which leads to cell cycle dysregulation and uncontrolled cell cycle progression. The E7 oncoprotein induces inhibition of the retinoblastoma protein, leading to activation of the transcription factor E2F. This causes the initiation of cell cycle progression and expression of p16, a surrogate marker for HPV infections. In contrast, HPV-negative tumors have typical p53 mutations [3]. The lower rate of p53 mutations in HPV-positive tumors can be explained by the suppression of p53 by the oncogene E6, by which these tumors do not need to select for p53 mutated cancer cells [4,5].

On the clinical level, patients with HPV-positive tumors have a significantly better overall prognosis, partly due to a higher radiosensitivity [6,7]. Until now, despite the evident differences, the selection of treatment modality has been based on the stage of disease and general condition of the patient, but not on HPV status. Standard therapy for patients with locally advanced HNSCC consists of radiotherapy, if indicated, in combination with chemotherapy and/or surgery [1]. However, the overall risk of disease recurrence for locally advanced HNSCC remains high, at about 40–60% [2]. Although various other treatments have been proposed, none seem to improve the overall survival of patients [8,9]. Therefore, further research into radiosensitizing treatment strategies is needed to improve the locoregional control and survival rates of locally advanced HNSCC, particularly for patients with HPV-negative tumors [10].

## 2. Influence of Hypoxia in Radiation Response of HNSCC

As previously mentioned, the better overall prognosis of HPV-positive HNSCC can, at least partly, be explained by the increased intrinsic radiosensitivity of these tumors [6,11,12,13]. This higher radiosensitivity is linked to impaired DNA repair mechanisms, cell cycle dysregulations, and p53 mediated apoptosis [7,14,15,16,17,18]. It has also been hypothesized that micro-environmental factors such as angiogenesis and the immune system can play a role in the differential radiotherapy response of HPV-positive and HPV-negative HNSCC. It has been postulated that HPV-positive HNSCC shows lower levels of VEGF expression, indicating an impaired angiogenesis in these cancers [19,20]. With the rising interest and crucial improvements in cancer immunotherapy, research about the immune system in tumors has also increased. In line with this, the higher immune response found in HPV-positive HNSCC could serve as a complementary explanation for the increased radiosensitivity [3,16,21]. However, the evidence for the latter is not substantial and includes only correlative studies demonstrating an increased rate of active immune cells in HPV-positive compared to HPV-negative HNSCC, suggesting a stronger antitumoral immune response in HPV-positive tumors [22,23,24,25].

Another key factor that is linked with the radiation response is hypoxia. Hypoxia is a common feature in solid tumors and is caused by a discrepancy between oxygen supply and demand, since the abnormal and inefficient tumor vasculature cannot fulfil the highly demanding proliferating tumor cells. Most tumors exhibit median oxygen levels <2%, while normal oxygen levels, i.e., normoxia, is defined as 20% oxygen [26]. Hypoxia can be chronic, due to prolonged limited oxygen diffusion, or acute. Acute hypoxia is typically alternated with reoxygenation, leading to cycling hypoxia. Both forms, chronic and acute hypoxia, exist simultaneously in a solid tumor with a heterogeneous distribution of the hypoxic areas [27,28,29]. Hypoxia is an overall negative prognostic factor, affecting both the treatment response and tumor biology. The importance of hypoxia as a cause of radioresistance was first observed by Gottwald Schwarz in 1909 [30,31]. Later, in 1953, Gray et al. described that the effectiveness of X-ray treatment might increase if the patients were breathing oxygen during irradiation [32]. This radiosensitizing effect of oxygen is attributable to the fixation of radical-induced DNA damage, also known as the ‘oxygen fixation theory’, making the damage permanent and irreversible [27,33,34]. Quantitively, this can be expressed by the oxygen enhancement ratio (OER), which is defined as the ratio of radiotherapy doses during low oxygen levels compared to doses in higher oxygen levels for the same biological effect [35,36].

The revelation of the importance of hypoxia in the radiation response, together with the knowledge of the higher radiosensitivity of HPV-positive HNSCC, resulted in studies investigating the link between the amount of hypoxia and HPV status. This hypothesis was further strengthened by a sub-analysis of a large, randomized trial testing the use of the hypoxia modifier nimorazole with radiotherapy in HNSCC (DAHANCA 5 trial) [37]. This study revealed that nimorazole was only beneficial in HPV-negative and not in HPV-positive tumors, suggesting a sort of interplay between hypoxia, HPV infections, and radiotherapy response [12]. A second DAHANCA 5 sub-analysis, in which a 15-gene hypoxia classifier was used to classify patients as having more or less hypoxic tumors, further confirmed the previous sub-analysis. The results showed that only the more hypoxic HPV-negative tumors benefited from the addition of nimorazole to radiotherapy, and outcomes were also not improved for either the less hypoxic or the HPV-positive tumors [38]. Conversely, numerous studies, including the sub-study of DAHANCA 5 by Toustroup et al. [38], demonstrated an equal degree of hypoxia and similar OERs in HPV-positive and HPV-negative HNSCCs [39,40,41]. Moreover, HPV-positive and HPV-negative cells display the same radioresistance under hypoxia and experience an equal sensitizing effect of nimorazole in vitro [41]. In vivo, a higher decrease in cell proliferation and hypoxic fraction after radiation was seen in HPV-positive compared to HPV-negative tumors [42]. Earlier in vitro findings, whereby a G2/M arrest was induced by irradiation of HPV-positive cells resulting in reduced cell proliferation and thus oxygen consumption, can explain this observation [3,7,14]. Both could be an additional explanation for the higher radiosensitivity of HPV-positive tumors and for the lack of benefit from hypoxic modifiers in these patients.

Overall, these findings suggest that the cellular and molecular mechanisms of hypoxia in HPV-positive and HPV-negative tumors are more important for the radiotherapy response than solely the oxygen fixation theory.

## 3. Cellular Effects of Hypoxia

Hypoxia does not merely modulate the radiation response, but additionally has various biological, molecular, and genomic effects on the tumor and micro-environment. The main goal of these hypoxia-induced effects is the survival and even proliferation of cancer cells in unfavorable hypoxic circumstances, leading to tumor progression, invasive growth, and cancer cell spreading [29]. In addition, hypoxia drives genetic instability through altered DNA repair and apoptosis resistance [28,33]. Clinically, these cellular changes lead to a more aggressive phenotype with higher recurrences, metastases, and poor prognosis. A major modulator of the adaptational response to hypoxia is hypoxia-inducible factor HIF-1. HIF-1 is a heterodimeric protein that consists of two subunits, HIF-1α and HIF-1β [29,43]. Its activity is regulated in an oxygen-dependent manner, more specifically by the oxygen-dependent degradation domain (ODDD) of HIF-1α [44,45,46,47]. In the presence of oxygen, rapid proteasomal degradation occurs through prolyl hydroxylation and subsequent binding to the von Hippel-Lindau (pVHL) tumor suppressor protein. In hypoxic circumstances, the rate of prolyl-hydroxylation decreases, whereby pVHL cannot bind HIF-1α and proteasomal degradation does not occur. By contrast, HIF-1α associates with HIF-1β after translocation to the nucleus, where it binds to hypoxia response elements (HRE) resulting in transcription of its target genes [33,48]. Additional pathways involved in the cellular response to hypoxia include the unfolded protein response (UPR) and kinase mammalian target of rapamycin (mTOR). Activation of the three pathways occurs independently, but with common processes downstream affecting angiogenesis, protein synthesis, metabolism, DNA repair, tumor immunity, and cell fate mechanisms [47,49,50,51]. In the next sections, we will discuss these hypoxia-induced cellular changes, for which a schematic overview is provided in Figure 1, and we will additionally point out the influence of HPV infections on these processes, which is summarized in Table 1.

### 3.1. Angiogenesis, Protein Synthesis and Metabolism

Broadly, the hypoxia-induced adaptational processes have two main objectives [63]. The first is to increase tumor oxygen levels, which can be achieved by inducing angiogenesis and erythropoiesis through the transcription of proangiogenic genes, such as vascular endothelial growth factor (VEGF) and erythropoietin (EPO) [64,65]. HPV infections also seem to influence these processes, since it has been shown that the HPV oncogene E7 increases HIF-1α levels and may influence angiogenesis [52,53,54], although studies also show an inverse correlation between angiogenic factors and HPV status [19,20].

As a second objective, the oxygen and energy consumption of cancer cells must be lowered. Since mRNA translation is one of the most consuming processes, the overall protein synthesis decreases during hypoxic stress as a way of energy conservation [28,66,67,68]. However, some mRNAs and proteins with specific functions (e.g., in survival, angiogenesis, hypoxic tolerance, and tumor growth) are conversely upregulated in hypoxic conditions to promote cancer cell survival and proliferation [63]. In HNSCC, differences in transcriptome and proteome have been described for HPV-positive and HPV-negative tumors [69,70,71].

Another attempt to lower the oxygen consumption involves metabolic transition to anaerobic glycolysis [49,63,72,73]. Normal cells prefer mitochondrial oxidation to produce energy, but in hypoxic circumstances they are forced to switch to inefficient glycolysis. In contrast, cancer cells always favor the more inefficient glycolysis for energy production, even when oxygen is plentiful. This process is referred to as aerobic glycolysis or the Warburg effect, and the rationale for it remains largely unclear [49,73]. However, the Warburg effect seems to contribute to tumor cell growth and proliferation through the production of multiple glycolytic intermediates that can be used in other biosynthetic pathways, like the pentose phosphate pathway [73,74]. In addition, the glycolytic metabolism of cancer cells is associated with radiation resistance, metastases, immunologic escape, and angiogenesis [73]. Interestingly, the metabolism in HNSCC differs between HPV-positive and HPV-negative tumors [22,55]. Numerous studies show that HPV-positive tumors tend to display higher rates of oxidative phosphorylation, while HPV-negative tumors are more glycolytic [22,56]. Moreover, a differential distribution of these metabolic pathways has been seen, with oxidative phosphorylation being predominantly present in the HPV-positive tumor core, while in HPV-negative tumors it was more active in the tumor periphery [22,55].

### 3.2. DNA Damage Response (DDR)

The DNA damage response is a complex overlapping network of genes that sense and repair DNA damage. The role of DDR genes, especially sensors such as ATM and ATR, and DNA repair pathways such as non-homologous end-joining (NHEJ) and homologous recombination (HR), in radiotherapy response has been established for several tumor types, including HNSCC. Moreover, as mentioned before, we and others have shown that HPV-negative and HPV-positive HNSCCs show differences in the DDR after radiotherapy. Hypoxia affects all DNA damage response pathways, in particular the mismatch repair (MMR) and HR pathways, by downregulating key genes in these pathways such as MSH1, MLH2, and RAD51 [33,75,76]. The influence of hypoxia and several DDR pathways has been extensively reviewed by Begg et al. [33]. Although several discrepancies have been reported about the influence of hypoxia on DDR, in general, hypoxia causes DNA damage and stress and initially activates DDR sensors, followed by the suppression of the DDR during prolonged hypoxia [33,76,77,78]. The suppressed DDR leads to increased mutagenesis and genetic instability, thereby contributing to the aggressive and malignant phenotype of hypoxic tumors [33,75,76,78,79].

Recent NGS studies [75,80] have shown that hypoxia is indeed associated with an increased mutational load across various cancer types [80]. In addition, they demonstrated that hypoxic tumors manifest characteristic driver-mutation signatures, such as TP53, MYC, and PTEN, suggesting that hypoxia applies a strong selective pressure to tumors. The characterization of such driver-mutation signatures is important, since it can contribute to the development of predictive biomarkers and targeted therapies for hypoxic tumors [75]. Moreover, a follow-up study of the same research group revealed that the combination of altered PTEN together with elevated hypoxia drives a polyclonal tumor architecture, resulting in poorer outcomes [80].

As mentioned earlier, the DDR differs between HPV-positive and HPV-negative HNSCC, with lower rates of HR and more error-prone NHEJ in HPV-positive tumors, resulting in genomic instability and radiosensitivity [14,18,57,58,59]. Hence, the effect of hypoxia on the DDR is also likely to differ between the two tumor types. However, this has not yet been investigated to our knowledge.

### 3.3. Immune Response

Tumors elicit complex immune responses, and hypoxia seems to play a significant role in this by promoting an immunosuppressive microenvironment through the recruitment and modulation of immune cells [47,81,82,83,84,85]. In addition, hypoxia seems to influence immune checkpoints and cause an upregulation and increased expression of PD-L1 through the HIF-pathway [47,50,86]. HPV infections play a role in shaping the intra-tumoral immune response, since correlative studies demonstrate a higher rate of active immune cells in HPV-positive compared to HPV-negative HNSCC, suggesting a stronger antitumoral immune response in HPV-positive tumors [22,23,24,25]. Hence, it is clear that both hypoxia and HPV infections interact with the intra-tumoral immune response; however, the exact mechanisms and interactions remain largely unknown. Further research is needed to clarify this, since these findings will help in the optimization of cancer immunotherapy.

### 3.4. Cell Death Mechanisms

Severe stress and/or unrepairable damage can induce various cell death mechanisms, such as apoptosis, autophagy, necrosis, necroptosis, and ferroptosis. Hypoxia has direct effects on these mechanisms and modulates the interactions between the different pathways [87]. Apoptosis, or programmed cell death, can be activated by several factors, with the p53 pathway as one of the key players. This makes p53 an important tumor suppressor protein that is mutated in 50–60% of human cancers, including HPV-negative HNSCC [61,62]. On the other hand, in HPV-positive HNSCC, p53 is generally not mutated, since it is downregulated by the HPV oncogene E6. Nevertheless, these tumors may still express some active p53 from the wild-type TP53 gene [60].

Not only HPV infections, but also hypoxia, seem to influence p53 function in cancer cells. However, studies report contradictory findings about the effects of hypoxia on apoptosis depending on the oxygen level [88]. Severe hypoxia and anoxia (0–0.5% O2) can induce apoptosis by the rapid activation of p53, while this does not occur in milder hypoxia (1–3% O2) [89,90,91,92,93]. Other studies report that HIF itself interacts with p53 since they both rely on the binding of p300/CBP for the initiation of transcription. This competition could lead to the suppression of p53 transcription and, thus, p53 function. Interestingly, after reoxygenation, a p53-dependent increase in apoptosis was observed. As expected, this was not the case for p53 mutated cells, resulting in genomic instability [33]. These findings indicate an existing interplay; however, the exact interactions between hypoxia and p53 and apoptosis remain largely unknown, and the results are ambiguous. Considering the typical p53 mutations and elevated levels of hypoxia in HPV-negative HNSCC, further investigations might be meaningful. Moreover, the possible interactions between the suppressed p53 pathway in HPV-positive tumors with hypoxia and apoptosis should be investigated [94].

## 4. Hypoxia-Targeting Strategies

In order to overcome the negative influence of hypoxia, various strategies to modify or target hypoxia in combination with RT have been extensively investigated in preclinical and clinical settings since the 1950s. Since most solid tumors are characterized by hypoxia, these trials investigating hypoxic modification and detection involved a variety of tumor sites, such as bladder, cervix, and lung. However, the head and neck area is one of the most investigated tumor sites regarding hypoxia [27]. An overview of the different hypoxia-targeting methods that have been tested in HNSCC is provided in Table 2.

The first hypoxic modification method that has been explored is hyperbaric oxygen (HBO) breathing, which leads to a physical increase of the blood oxygen levels [27,103,104]. Some of these early trials demonstrated a significant improvement of local control and overall survival, especially in head and neck tumors [95]. However, the use of HBO also leads to a significant increase in normal tissue side effects, negating the outcome benefits. In addition, HBO is a technically complicated approach with many practical difficulties [104]. Since the disadvantages of this radiosensitizing method outweigh the advantages, it was never implemented in daily clinical practice. Based on the principle of HBO, a method to increase oxygen delivery through the blood, the ARCON strategy, arose. ARCON involves a combination of carbogen, nicotinamide (vitamin B6 analog), and accelerated radiotherapy [105]. A randomized phase III trial showed a significant gain in regional control and similar toxicities compared to accelerated radiotherapy in advanced laryngeal cancer. However, it should be noted that the radiotherapy doses were lower in the ARCON arm because of the high rates of larynx necrosis in previous phase II trials [106,107]. Interestingly, the control benefit was only observed in tumors with a high hypoxic fraction, defined by pimonidazole staining, demonstrating the importance of patient stratification [98].

Other hypoxic modification methods are based on hypoxia-activated prodrugs (HAP), which are selectively activated in hypoxic circumstances by enzymatic reduction [108,109,110]. Tirapazamine (TPZ) causes DNA strand breaks after hypoxic activation, but was not shown to be beneficial when added to chemoradiotherapy [96]. However, after patient stratification with ^18^F-MISO PET, treatment with TPZ did decrease the ratio of recurrences in the hypoxic subpopulation [97]. Another DNA-damaging HAP is evofosfamide (TH-302), with discouraging results in two phase III trials investigating its effect in sarcoma and pancreatic carcinoma [108,111,112]. Preclinically, the HAP PR-104 showed initially promising results with hypoxia-selective activation and enhanced antitumor effects [99]. However, further research revealed an additional activation mechanism independent of oxygen by the reductase AKR1C3, resulting in a dose-limiting myelotoxicity clinically [100,101,113]. Based on PR-104, a new and improved HAP was developed: CP-506. The main advantages of this novel agent include AKR1C3 resistance, water-solubility, orally bioavailability, and a large bystander effect [102]. The bystander effect results from local diffusion of the active drug metabolites, and is important to overcome unequal delivery of the drug [114]. These favorable pharmacokinetic properties were confirmed in vitro and in vivo, next to a broad antitumor activity in HNSCC [102].

Up till now, the oxygen mimetic nimorazole is the only hypoxic radiosensitizer that has been implemented in clinical practice, since a large phase III trial (DAHANCA 5) showed a significant outcome benefit by combining radiotherapy with nimorazole in HNSCC [3,37]. However, this regimen is only standard care in Denmark, as up until now, no further trials have demonstrated a benefit of adding nimorazole to current (chemo)radiotherapy schedules. Recently, early results of a randomized phase III trial testing the combination of nimorazole with accelerated chemoradiotherapy in HPV-negative HNSCC were presented. At 2 years, the locoregional control probability was not clinically different between the two arms, either in the entire population (63.8% for nimorazole and 72.1% for placebo), or in the hypoxic gene-positive patients [115]. Currently, another randomized trial is ongoing to confirm the DAHANCA results (NIMRAD, NCT01950689). In this placebo-controlled trial, the addition of nimorazole to radiotherapy is compared to radiotherapy alone in patients with locally advanced HNSCC not suitable for synchronous chemotherapy or cetuximab. The study is not actively recruiting anymore, but no results have been published yet.

As summarized above, only one clinical trial [37] could demonstrate a significant outcome benefit and better radiation response with hypoxia modification. A major obstacle in these trials has been the biological variety and heterogeneity of tumor hypoxia [27,109,116,117], underlining the need for proper patient stratification through accurate hypoxia detection [33,117]. In addition, it should be mentioned that studies comparing the impact of hypoxia on radiation response, with or without hypoxic modification, in HPV-positive and HPV-negative tumors are scarce, and the results of various analyses are contradictory, leading to a series of uncertainties [12,38,39,41,42]. Further studies are warranted to clarify the underlying biological processes involved in the response to hypoxia and radiotherapy, and to determine the possible benefits of hypoxic modification in HNSCC.

## 5. Detection of Hypoxia

Tumors display different degrees and distributions of hypoxia, leading to variable responses to hypoxia-targeting strategies. Therefore, information about the extent of hypoxia is crucial in order to select patients who will likely benefit from hypoxic modification. In addition, pretreatment information about the hypoxic status has proven prognostic value [3,118]. Since the importance of patient stratification has become clear, various approaches for the feasible and reliable detection of tumor hypoxia have been investigated [3,27,35,118,119].

The first and the oldest detection method includes the direct measurement of tumor oxygen concentration using polarographic needle electrodes, for example the Eppendorf electrode. The key advantage of this method lies in the fact that it measures oxygen levels directly and that it has a proven correlation with prognosis in multiple cancer types, including HNSCC [118,120,121,122,123,124,125,126]. By contrast, the invasiveness of the procedure and the inaccessibility of most tumors are important weaknesses. In addition, the electrodes are unable to differentiate between hypoxia and necrosis, and they do not provide spatial resolution [3,118,119]. These disadvantages spurred the development of more user-friendly and versatile detection methods.

Later techniques are based on detecting and monitoring hypoxia markers in biopsies or resection specimens. These markers can be endogenous, like hypoxia-inducible factor 1α (HIF-1α), carbonic anhydrase IX (CAIX), glucose transporter 1 (GLUT-1), and vascular endothelial growth factor (VEGF) [119]. Overexpression of these markers can be linked to poor prognosis in HNSCC and other tumor types; however, this correlation is not always specific [127,128,129,130,131,132,133,134,135,136]. Hypoxia detection with exogeneous markers uses non-physiologic substances, such as pimonidazole and EF5, that accumulate in hypoxic areas by chemical reduction and covalent binding to macromolecules. The markers are administered to the patient (intravenously or per os), whereafter a tissue biopsy is taken to perform immunohistochemistry or -fluorescence, thereby visualizing the hypoxic areas with an excellent spatial resolution [3,119]. The prognostic value of exogenous hypoxia markers has been demonstrated in HNSCC in a sub-study of a phase II ARCON trial. The results revealed that tumors with high-level pimonidazole binding had lower locoregional control rates compared with less hypoxic tumors [137]. Furthermore, a translational side study of a phase III trial investigating ARCON in laryngeal cancer observed a significantly improved regional control with the combination treatment in tumors with high hypoxic fraction, determined with pimonidazole, but not in low hypoxic tumors [98].

A third biopsy-based detection strategy assesses gene expression signatures, which consist of a selection of genes specifically upregulated under hypoxia. This upregulation is a result of the cellular adaptation and transcriptional response to hypoxia, mainly regulated by the HIF pathway [138,139,140]. The feasibility of hypoxia gene expression signatures as a prognostic tool in HNSCC patients has been proven in multiple studies [38,141,142,143]. Toustroup et al. showed inferior outcomes after radiotherapy in patients with hypoxic head and neck tumors, as determined by a 15-gene hypoxia classifier. Moreover, only in the more hypoxic tumors was the use of the hypoxic modifier nimorazole significantly beneficial [38]. Another trial using a 26-gene hypoxia signature demonstrated comparable results [143]. The prognostic value of these hypoxia signatures in HNSCC was later confirmed in a comparative analysis by Tawk et al. [142]. In contrast, in a recent paper of our research group in which we evaluated the use of the 15-gene hypoxia classifier in patients with oropharyngeal cancer treated with accelerated chemoradiotherapy, the prognostic value of the classifier could not be validated [144].

Despite their benefits, both the electrode- and tissue-based measurements are less user friendly, since they involve invasive techniques and only inform on oxygen levels at a specific time point, unless repeated samples are taken. This makes them less suitable for monitoring tumor oxygenation and/or treatment response, underlining the need for non-invasive approaches like hypoxia imaging [3].

Positron emission tomography (PET) with hypoxia-selective tracers is more useful in clinical practice, allows hypoxia monitoring during treatment, and offers a reliable and reproducible visualization of tumor oxygenation, as demonstrated in various tumor types [145,146,147]. In addition, hypoxia PET can facilitate adaptive radiotherapy, by which a higher dose is prescribed to the hypoxic radioresistant sub-volumes of the tumor [118]. Several randomized phase II and phase III trials are testing this so called ‘dose painting and dose escalation’ with hypoxia PET in HNSCC (NCT02089204, NCT02352792, NCT01212354) [148]. The preliminary results of 25 patients with oropharyngeal and hypopharyngeal carcinoma show 2-year locoregional control rates in favor of dose escalation [149]. In addition, in nasopharyngeal carcinoma, the first results of dose escalation with hypoxia PET are promising, with improved local control rates compared to conventional chemoradiotherapy [150]. However, this strategy remains challenging, since PET is not ideal for the accurate determination and delineation of the tumor borders and volume [151]. In addition, it is unclear what radiation doses are needed for optimal tumor control of the hypoxic subregions. Moreover, these regions and the amount of hypoxia fluctuate during radiotherapy, for which adaptive planning should be necessary. However, the optimal frequency and timing of PET evaluations and subsequent plan adaptations during treatment remain unknown [148,152].

Various hypoxia PET tracers have been developed, but ^18^fluoromisonidazole ^(18^F-MISO), which consists of labeled 2-nitroimidazoles, which selectively bind to macromolecules in hypoxic cells, is the one most commonly used and investigated [118,148]. In a sub-study of the RTOG 98.02 trial [153], the hypoxia status in patients with HNSCC was assessed using ^18^F-MISO PET. In this study, patients with hypoxic tumors had a higher risk of locoregional recurrences, demonstrating the prognostic value of ^18^F-MISO PET. As mentioned before, the addition of tirapazamine to chemoradiation significantly improved the outcome for these patients [97].

A more novel hypoxia-selective PET tracer is ^18^F-fluoroazomycin arabinoside (FAZA), with promising prognostic potential for hypoxia detection in HNSCC as well [40]. Other new PET tracers that have recently been introduced in the clinic include ^18^F-FETNIM, ^18^F-EF3, ^18^F-EF5, and ^18^F-HX4 [148,154,155]. More indirect PET-based detection methods rely on the quantification of blood flow or oxygenation-dependent flux changes through ATP-generating pathways, for example anaerobic glycolysis [118].

The current techniques to detect hypoxia show promising results regarding their prognostic and predictive value; however, each has its inherent weaknesses. Further validation of these techniques is warranted in different tumor types and treatment regimens. Additionally, the ideal timing for hypoxia detection still remains a subject for debate. The focus of future research should lie on patient selection with reliable hypoxia detection methods in combination with testing of hypoxia-targeting agents in the hypoxic subgroup.

## 6. Conclusions

HPV-positive and HPV-negative HNSCCs differ at the biological and clinical level, with a better overall prognosis for the HPV-positive group. This improved prognosis is, at least partly, attributable to the increased radiosensitivity and thus better radiation response of HPV-positive tumors. Besides HPV infections, hypoxia is also an important modulator of the radiation response. The most well-known explanation for this interplay lies in the ‘oxygen fixation theory’, which arose many decades ago. Nevertheless, more recently, it has become clear that hypoxia induces a variety of cellular processes that also, and probably to a more important extent, seem to influence the therapy response. Moreover, these cellular changes affect the biological behavior and phenotype of hypoxic cancer cells. In addition, recent NGS studies show that hypoxia causes genetic instability, which can be linked to a poorer prognosis. However, it remains unclear whether and how these cellular and genetic alterations influence the therapy response and the increased radiosensitivity of certain tumor types like HPV-positive HNSCC. Further studies investigating the fate of hypoxic cells and the hypoxia-induced cellular adaptions and genetic alterations, with and without radiotherapy, can shed light on the importance and interplay of these various known biological processes.

Because of its negative impact on tumor responsiveness and overall prognosis, hypoxia remains an important therapeutic target to be exploited. Unfortunately, up till now, the clinical successes of hypoxia-targeting strategies are poor. Some recent HAPs show improved targeting abilities in several preclinical trials; however, the responses are still heterogeneous, underlining the need for proper hypoxia stratification. The existing detection methods are promising, but need further validation in prospective randomized trials. Ideally, these should be combined with the testing of novel hypoxia-targeting strategies.

## Figures and Tables

**Figure 1 cancers-13-05959-f001:**
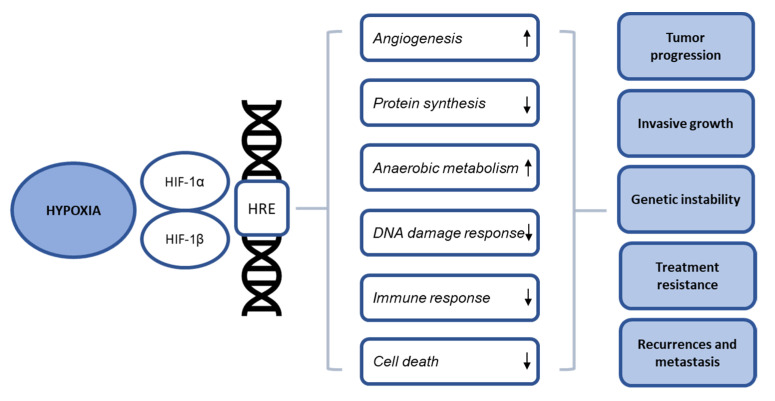
Schematic summary of the cellular effects of hypoxia. HIF-1: Hypoxia inducible factor-1, HRE: Hypoxia response elements.

**Table 1 cancers-13-05959-t001:** Overview of differences in cellular processes between HPV-positive and HPV-negative HNSCC.

Cellular Processes	HPV-Positive HNSCC	HPV-Negative HNSCC
Angiogenesis	Increased levels of HIF-1α [52,53,54] Inverse correlation with angiogenic factors [19,20]	Higher levels of angiogenic factors [19,20]
Metabolism	Higher rates of oxidative phosphorylation, mostly in the tumor core [22,55,56]	Higher rates of glycolysis, mostly in the tumor core [22,55,56]
DNA Damage Response	Impaired DNA DSB repair with less HR and more NHEJ [14,18,57,58,59]	Enhanced DNA DSB repair [14,18,57,58,59]
Immune Response	Higher rate of active immune cells [22,23,24,25]	Lower rate of active immune cells [22,23,24,25]
Cell Death Mechanisms	p53 suppression by HPV oncogene E6 [4,5,60]	p53 mutations [3,61,62]

HIF-1α: Hypoxia inducible factor-1α, DNA DSB: DNA double strand breaks, HR: Homologous recombination, NHEJ: Non-homologous end-joining.

**Table 2 cancers-13-05959-t002:** Overview of hypoxia targeting strategies investigated in HNSCC.

Hypoxia- Targeting Strategy	HNSCC Trials	Treatment Schedule	Hypoxia Detection Method	Outcome	Toxicity
HBO	Overview by Overgaard [95]	RT with HBO or RT alone	/	Improved local control (*p* = 0.003)	Increased normal tissue toxicity
TPZ	RTOG 98.0 (phase II) [96,97]	Chemo-RT with TPZ or chemo-boost	^18^F-MISO PET	Hypoxic tumors improved locoregional control (*p* = 0.015)	More febrile neutropenia and grade 3 or 4 late mucous membrane toxicity
ARCON	Janssens et al. (phase III) [98]	ARCON or accelerated RT alone	Pimonidazole (exogeneous marker)	Hypoxic tumors improved regional control (*p* = 0.04)	Similar (however lower RT dose in ARCON arm
Nimorazole	DAHANCA 5 (phase III) [38]	RT with nimorazole or placebo	15-gene hypoxia classifier	HPV-negative hypoxic tumors improved locoregional control (*p* = 0.002)	Minor nausea and vomiting
PR-104	Preclinical data [99,100,101]	/	/	Selective activation and enhanced antitumor effects	Dose-limiting myelotoxicity
CP-506	Preclinical data [102]	/	/	Favorable pharmacokinetics and broad antitumor activity	/

RT: Radiotherapy, HBO: Hyperbaric oxygenation, TPZ: Tirapazamine.

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
