# Peer review of "Hypoxia and Its Influence on Radiotherapy Response of HPV-Positive and HPV-Negative Head and Neck Cancer"

_cancers, 2021, doi:10.3390/cancers13235959_

Round 1

Reviewer 1 Report

Wegge et al. provide an overview over the previous work on hypoxia and its influence on radiotherapy of HNSCC. They cover both current treatment as well as detection strategies for hypoxia.

My comments are mostly in regards to English, mainly regarding the lack of articles in several places. Two examples in line 43 and 45: "On biological level" should be "on the biological level", "action of two main oncoproteins" should the "action of the two main oncoproteins". This occurs several times throughout the manuscript.

In line 37, the authors claim HNSCC is caused "by either alcohol and/or nicotine or". I think this should be tobacco instead of nicotine unless the authors want to get into a long discussion with the field about the carcinogenic nature of nicotine.

Line 39: "The incidence of HPV-positive tumors, one the other hand, is increasing." This sentence lacks a classifier in the previous sentence as it's not stated whether HPV-negative tumor cases are decreasing or constant. This is important information for people not familiar with HNSCC.

It might be good if the authors provide the definition for hypoxia (versus normoxia; how much O2) in the second section to familiarize people with the matter.

In section 3.3:

Tumors do not have a complex immune response but they are the target of the immune system. They might harbor a complex infiltrate of immune cells or elicit complex immune responses. I would also rephrase the sentence "Next to hypoxia, HPV-infections .... " to "HPV-infections play a role in shaping the intratumoral immune response". Also in line 223 it should read "interact with the intratumoral immune response".

Author Response

Point 1: My comments are mostly in regards to English, mainly regarding the lack of articles in several places. Two examples in line 43 and 45: "On biological level" should be "on the biological level", "action of two main oncoproteins" should the "action of the two main oncoproteins". This occurs several times throughout the manuscript.

Response 1: We added articles in the described places in line 43 and 45. In addition, we added articles in lines 53, 177, 194, 267, 296, 333, 337 and 374.

Point 2: In line 37, the authors claim HNSCC is caused "by either alcohol and/or nicotine or". I think this should be tobacco instead of nicotine unless the authors want to get into a long discussion with the field about the carcinogenic nature of nicotine.

Response 2: We changed nicotine into tobacco in line 37.

Point 3: Line 39: "The incidence of HPV-positive tumors, one the other hand, is increasing." This sentence lacks a classifier in the previous sentence as it's not stated whether HPV-negative tumor cases are decreasing or constant. This is important information for people not familiar with HNSCC.

Response 3: Worldwide, the incidence of HNSCC is increasing, as stated in line 35. The majority of these tumors are HPV-negative, however the incidence of HPV-positive tumors is increasing. Hence, the incidence of both groups is increasing.

Point 4: It might be good if the authors provide the definition for hypoxia (versus normoxia; how much O2) in the second section to familiarize people with the matter.

Response 4: We added extra information about the levels of oxygen in tumors and normal oxygen levels i.e. normoxia in line 83-84.

Point 5: In section 3.3: Tumors do not have a complex immune response but they are the target of the immune system. They might harbor a complex infiltrate of immune cells or elicit complex immune responses. I would also rephrase the sentence "Next to hypoxia, HPV-infections .... " to "HPV-infections play a role in shaping the intratumoral immune response". Also in line 223 it should read "interact with the intratumoral immune response".

Response 5: We adjusted these sentences as suggested in line 218, 223 and 227.

Reviewer 2 Report

A very interesting narrative review exploring the role of hypoxia in radiotherapy to treat HNSCC. although I am not a radiotherapist, I found the article very informative, and eligible to be published after minor revisions:

What do you think about the use of nimorazole in radiotherapy to treat HNSCC? Do you think it should be implemented also in other countries?

I think that Fmiso PET should be further discussed, as it may become a mainstay in controlling the evolution of HNSCC

Page 2 line 59 you should add: "although various other treatments habìve been proposed, none of them seem to improve the overall survival of patients" and cite an article such as: doi: 10.3390/medicina57060563. and doi: 10.3390/curroncol28040213.

Author Response

Point 1: What do you think about the use of nimorazole in radiotherapy to treat HNSCC? Do you think it should be implemented also in other countries?

Response 1: Since the benefit of nimorazole with (chemo)radiotherapy could not be demonstrated in other trials than the DAHANCA 5 trial, we do not think that it should be implemented in other countries. The preliminary negative results from a recent randomized phase 3 trial (DAHANCA 29-EORTC 1219) support this notion. Further large prospective trials investigating nimorazole with chemoradiotherapy together with proper patient stratification are needed before nimorazole should be implemented worldwide.

Point 2: I think that Fmiso PET should be further discussed, as it may become a mainstay in controlling the evolution of HNSCC

Response 2: We have added a section (line 384-396) about 18F-MISO PET being used in dose painting and dose escalation trials, since this is a recent development in radiation oncology with a strong growth potential. In this section, we describe current clinical trials with their preliminary results and the limitations of this new radiation strategy.

Point 3: Page 2 line 59 you should add: "although various other treatments habìve been proposed, none of them seem to improve the overall survival of patients" and cite an article such as: doi: 10.3390/medicina57060563. and doi: 10.3390/curroncol28040213.

Response 3: We added the sentence as suggested in line 59.